# Wave Signatures in Total Electron Content Variations: Filtering Problems

**Boris Maletckii [1,2,]\*** , **Yury Yasyukevich [1]** and **Artem Vesnin [1]**

[1]  Institute of Solar Terrestrial Physics of Siberian Branch of Russian Academy of Sciences (ISTP SB RAS), st. Lermontov 126a, 664033 Irkutsk, Russia; yasukevich@iszf.irk.ru (Y.Y.); artem_vesnin@iszf.irk.ru (A.V.)

[2]  Faculty of Physics, Irkutsk State University, Gagarin Boulevard, 20, 664003 Irkutsk, Russia

\*  Correspondence: maletskiy@iszf.irk.ru; Tel.: +7-924-607-5754

**Abstract:** Over recent years, global navigation satellite systems (GNSSs) have been increasingly used to study near-Earth space. The basis for such studies is the total electron content (TEC) data. Standard procedures for detecting TEC wave signatures include variation selection and detrending. Our experience showed that the inaccurate procedure causes artifacts in datasets which might affect data interpretation, particularly in automated processing. We analyzed the features of various detrending and variation selection methods. We split the problem of the GNSS data filtering into two subproblems: detrending and variation selection. We examined centered moving average, centered moving median, 6th-order polynomial, Hodrick–Prescott filter, L1 filter, cubic smoothing spline, double-applied moving average for the GNSS-TEC detrending problem, and centered moving average, centered moving median, Butterworth filter, type I Chebyshev filter for the GNSS-TEC variation selection problem in this paper. We carried out the analysis based on both model and experimental data. Modeling was based on simple analytical models as well as the International Reference Ionosphere. Analysis of TEC variations of 2–10 min, 10–20 min, and 20–60 min under insufficient detrending conditions showed that the higher errors appear for the longer periods (20–60 min). For the detrending problem, the smoothing cubic spline provided the best results among the algorithms discussed in this paper. The spline detrending featured the minimal value of the mean bias error (MBE) and the root-mean-square error (RMSE), as well as high correlation coefficient. The 6th-order polynomial also produced good results. Spline detrending does not introduce a RMSE more than 0.25 TECU and MBE > 0.35 TECU for IRI trends, while, for the 6th-order polynomial, these errors can exceed 1 TECU in some cases. However, in 95% of observations the RMSE and MBE do not exceed 0.05 TECU. For the variation selection, the centered moving average filter showed the best performance among the algorithms discussed in this paper.

**Keywords:** ionosphere; total electron content; global navigation satellite system; data filtering; data processing

## 1. Introduction

In recent years, global navigation satellite systems (GNSSs) have been increasingly used to study the near-Earth space, including storm patterns [1], solar activity impact [2], travelling ionosphere disturbances (TID) features [3], and real-time monitoring of the TID [4] and multi-TID [5]. The basis for such studies is the total electron content (TEC) data. TEC is obtained from dual-frequency measurements of phase and pseudo-range (group). To study traveling ionospheric disturbances (TIDs), phase TEC is usually used, since its noise level is much lower than that of pseudo-range measurements.

One of the important features of the TEC data is the trend caused by the motion of satellites located in medium Earth orbits. Standard procedures for detecting TEC variations include filtration

and detrending. In the bulk of studies, the following techniques are used to detrend data: the moving average [6,7], and high-order polynomial [8–10]. To filter data, one most often uses either the moving average [3], or the Butterworth filter [11,12] used a short-time Fourier transform (STFT) to avoid the filtering problem. Unfortunately, the STFT approach cannot always be used for several algorithms applied now, such as interferometry [13], or solar flare effect detection [14]. Our own experience showed that the artifacts originating from inaccurate filtering might impact upon processing results interpretation, particularly at automatic processing, such as SIMuRG [15]. Therefore, correct filtration is a critical for GNSS automatic data treatment techniques.

Herewith, we should note that the filtration characteristics have not been practically discussed in the literature. To the authors' knowledge, no publication can be found that addresses the issue of comparing different variation selection and detrending approaches.

The TEC series after correct filtration should at least meet the following criteria: 1. Have zero mean; 2. Preserve the relative amplitudes of spectral components in the filtered band; 3. Preserve the spectral composition only for the filtered band; 4. Preserve the phase of components.

Figure 1 shows how the problem manifests itself when the aforementioned criteria are not satisfied. The data are provided for IRKJ (52.22 N, 104.31 E) and BADG (51.77 N, 102.23 E) stations of the IGS [16] network and satellite G06 on 14 August 2018. Panels (a) and (b) show original TEC series for IRKJ and BADG stations, correspondingly. The TEC trend based on 60-min centered moving average (CMA) is shown in red. Panels (c) and (d) show 60-min-CMA-detrended TEC series for the same stations. Panel (e) show 20–60 min TEC variations, when the detrended TEC series is smoothed by 20-min CMA. After this is done, it is conventionally assumed no components with periods beyond 20–60 min are left in the series, which is not the case as we can see. Due to an insufficiently complete filtration, the data has uncharacteristic negative values. Moreover, TEC variation series contain long-term variation with a period beyond filtration band.

This artifact appears for processing with the centered moving average. If the filtration was correct, we would record both positive and negative values ("zero mean" condition). Panel (f) shows cross-correlation functions between TEC variation series for IRKJ and BADG. Observed long-term trend ("preserve spectral composition only for the filtered band" condition is also broken) results in broadening of the maximum of cross-correlation function, preventing correct estimation of the wave propagation delay. Further automatic processing (such as TID interferometry) will suffer from such an artifact. It is difficult to say which maximum (left or right) in Figure 1f corresponds to the correct delay.

Figure 2 shows how the artifacts from Figure 1 appear spatially when mapping 20–60 min TEC variations on the Earth's surface. These and similar figures (not shown here) reveal preferable negative values, which does not correspond to basic physical considerations. Thus, it is impossible to correctly interpret the data, as well as to use it for further application. For example, computer vision techniques for irregularity detection. A large number of negative values are caused by the chosen filtration procedure does not meet all requirements. This indicates that the problem shown in Figure 1 is common and does not depend on site selection.

In this study, we analyze the features of various detrending and filtration techniques. Each of these subproblems (detrending and filtration) is addressed separately. This allows us to determine effects of each step on the results of preliminary treatment of TEC series. We address the problem of detrending separately since a trend is determined by different basis (for example, polynomial expansion) than variations (harmonics decomposition). The validation is made via modeling when we define the input data and can directly estimate the error. Experimental data analysis finalizes this study.

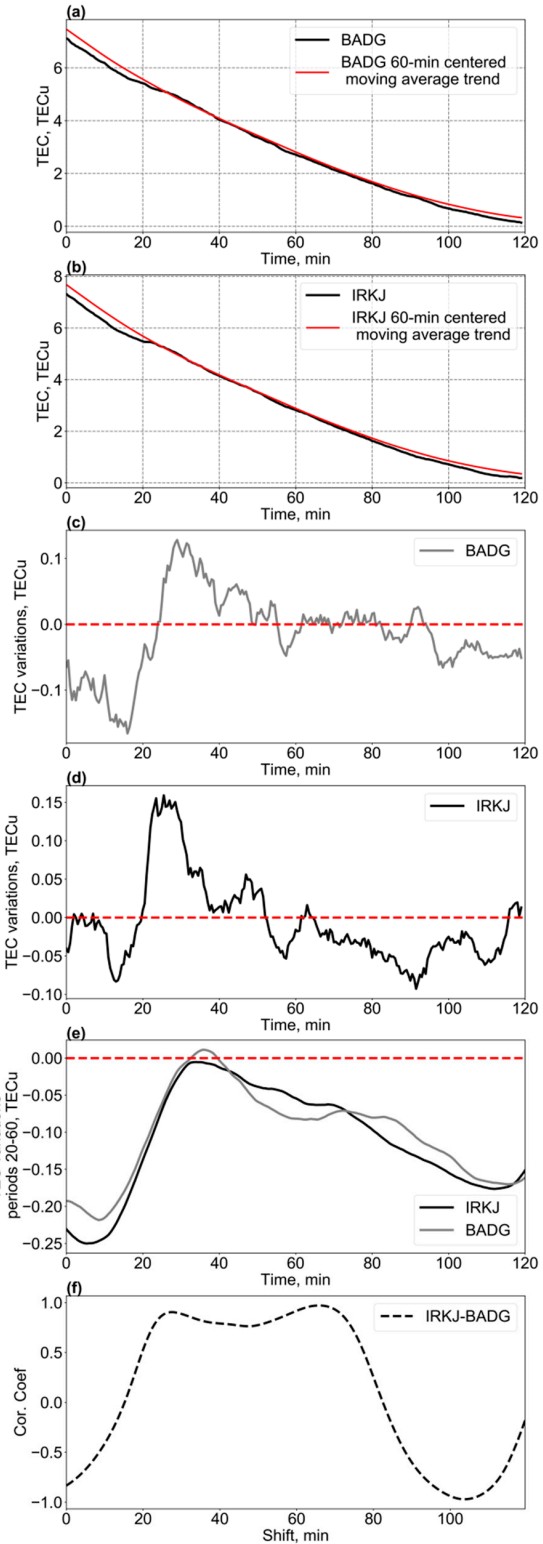

**Figure 1.** Effects of inaccurate filtration on subsequent data analyses: original (black lines) TEC series and 60-min CMA TEC trend (red lines) for IRKJ (**a**) and BADG (**b**) stations; 60-min-CMA-detrended TEC series for IRKJ (**c**) and BADG (**d**); 20–60 min TEC variations (**e**) for IRKJ (black line) and BADG (gray line); Cross-correlation functions (**f**) between TEC variation series for IRKJ and BADG. All original TEC series are from G06 satellite signal. Red dashed lines on panel (**c**–**e**) show the zero-level.

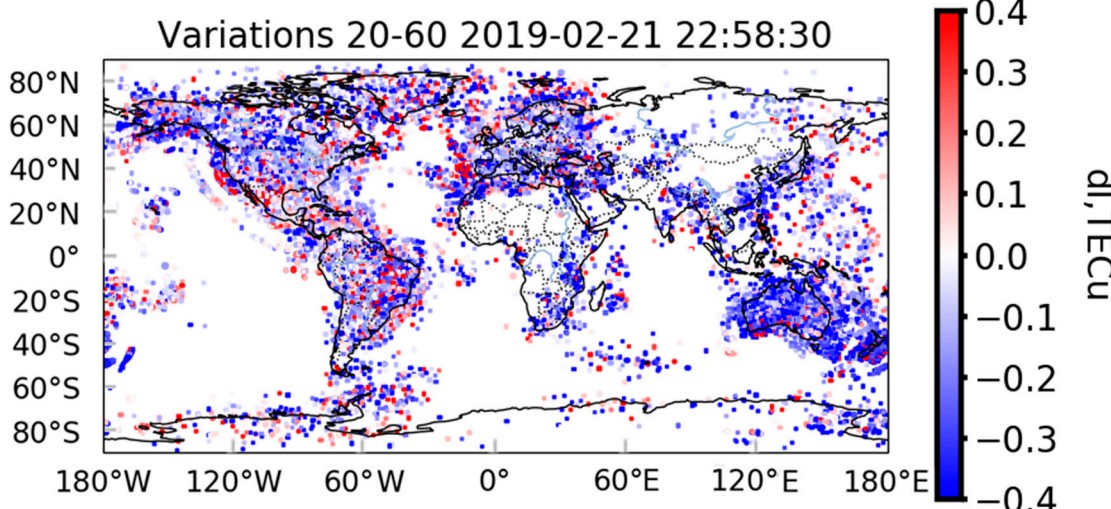

**Figure 2.** Map of 20–60 min TEC variation. 21 February 2019. 22:58:30 UT.

## 2. Methodology

We reviewed basic algorithms of this class of problems. We used the following techniques to detrend TEC data:

1. Centered moving average (CMA).
2. Centered moving median (Median).
3. 6th-order polynomial (Polynom).
4. Hodrick–Prescott Filter (Hod-Pres). The input data for this filter is λ, which determines trend flattening. This value should be selected as the fourth degree for the rate of the frequency change in the signal that should be obtained upon detrending [17]. In our case, λ = ~129,600.
5. L1 Filter (l1). The input data for this filter is λ, which defines trend smoothing. We have never found explicit instructions for how to determine this frequency-dependent parameter. Therefore, its value (λ = 0,5) was found by modeling.
6. Cubic smoothing spline (Spline). The smoothing spline input parameter is Smoothing Factor, which determines trend smoothing. The Smoothing Factor value (8) was found empirically.
7. Double use of the centered moving average (Double CMA).

We are interested in the following ranges for data filtering: 2–10 min (corresponding to the AGW-oscillation branch), 10–20 min (corresponding to medium-scale TID periods), and 20–60 min (corresponding to large-scale TID periods). For variation selection, we used:

1. Centered moving average (CMA).
2. Centered moving median (Median).
3. Butterworth filter of 8th order (Butter).
4. Type I Chebyshev filter of 8th order (ChebyI).

To analyze the quality of using various algorithms, we used the following parameters.

1. Mean bias error (MBE) $M_{RF}$:

$$M_{RF}(\Delta I) = \frac{1}{N} \sum_{i=1}^{N} (I_R - I_F),\qquad(1)$$

where $I_R$, $I_F$ are the reference signal and the signal obtained upon implementing procedures; $N$ is the number of data points.

2. Root-mean-square error (RMSE) σ, i.e., the standard deviation of the residuals between the modeled and the recovered signals:

$$\sigma_{RF} = \sqrt{\frac{1}{N} \sum_{i=1}^{N} (I_R - I_F)^2}, \tag{2}$$

3. Correlation coefficient $K$ between the known used signal $I_R$ and the recovered signal $I_F$ upon implementing filtering/detrending procedures:

$$K_{RF} = \frac{\sum_{i=1}^{N} (I_R - \langle I_R \rangle)_i (I_F - \langle I_F \rangle)_i}{\sqrt{\sum_{i=1}^{N} (I_R - I_R)_i^2 \sum_{i=1}^{N} (I_F - I_F)_i^2}}, \tag{3}$$

The calculation performance is another key point when automatically process a great volume of data. The number of stations is several thousand since the early 2000s and it is continuously growing. The development of satellite constellations enables to simultaneously observe (by now) more than 40 satellites of various systems.

## 3. Modeling Results

The TEC time series are assumed to be additive combination of a trend and a signal (TEC variations). For variation selection subtask we also add an error (Gassian noise). We modeled the above three period bands (2–10, 10–20, 20–60 min). As we show below, maximal errors were observed for 20–60 min. Thereupon, the validation presented in this paper is provided for this range and for others assumed to be held.

### 3.1. Detrending

To analyze the detrending quality, we generate reference signal $I_R$ in the form of a wave packet comprising the frequencies within the bands of interest. Then the trend is generated and the wave packet is superimposed. The resulting series is detrended using each of the above-described techniques to obtain a filtered (detrended) signal. Finally we analyze the $K_{RF}$, $\sigma_{RF}$, and $M_{RF}$ parameters to estimate whether the technique meets the requirements or not.

Figure 3 presents the signal model (top) and the model for the trend signal (bottom). As a part of the wave packet, there are three harmonics with periods of 20-, 40-, and 60-min. The harmonic amplitude, A = 0.2 TECU. The model for the reference signal looks like:

$$I_R = A \cdot e^{-\frac{(t - t_m)^2}{2d_t^2}} \cdot \sum_{i}^{n} sin(\omega_i t), \tag{4}$$

where the $t_m$ parameter determines the position of the wave packet envelope maximum, $d_t$ is half-width of the envelope (for all the temporal oscillation ranges, it equals 50 min). The trend model corresponds to typical TEC trends and is described by equation:

$$Trend(t) = B \cdot |t - t_0|^3, \tag{5}$$

where the $t_0$ parameter determines the position of the trend minimum and is equal 250 min (half of the array length), $B$ is the amplitude (TECu/minutes) of the trend and equal to $62/(250^3)$.

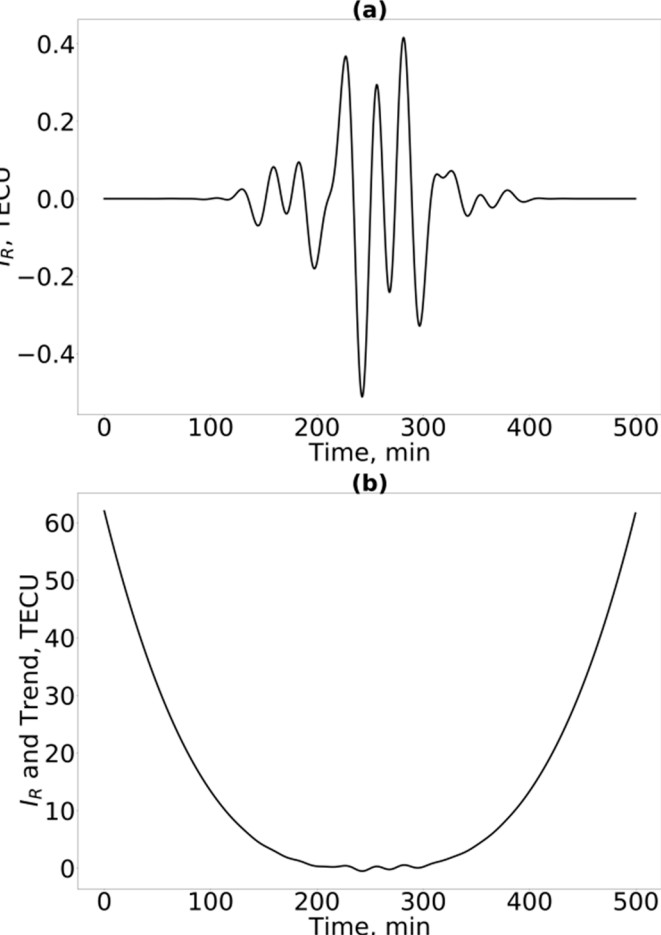

**Figure 3.** Model for the $I_R$ signal (**a**) and the trended signal (**b**). On the bottom panel signal is in the middle (minimum) of the trend.

When CMA detrending for the case presented in Figure 3 (the signal on the top panel is in the middle of the observation interval, i.e., in the trend minimum), the mean bias error ($M_{RF}$) is 0.106 TECU (20–60 min). For the 2–10 and 10–20 min signals, the $M_{RF}$ equals 0.004 TECU and 0.013 TECU, respectively. Since detrending for 20–60 min band demonstrates the worst effect, below we show figures for this band, while calculations were performed for all three bands.

One may expect that the detrending quality depends on the trend change rate. For the analysis, we simulated the series with various wave packet position relative to trend minimum. However, we did not find such a dependence for $M_{RF}$ and $\sigma_{RF}$. Figure 4 shows the mean bias error, $M_{RF}$, which was averaged for different (against the trend minimum) the wave packet position. From Figure 4, we can see that the moving average, double filtering by the moving average, and the moving median introduce an essential constant component to the recovered signal $I_F$. The value of this constant component is commensurable with the amplitudes of the initial wave-packet harmonics, which is a serious artifact and may affect the further processing. The remaining detrending techniques introduce a smaller constant component equal to the tenths and the hundredths (cubic spline, Hodrick–Prescott filter, L1 filter, 6th-order polynomial) of the input amplitude.

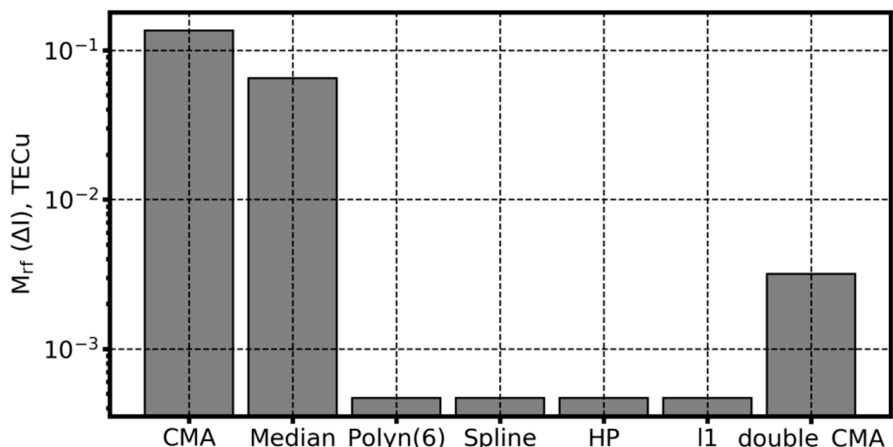

**Figure 4.** The mean bias error, $M_{RF}$ for various detrending techniques. Y-axis has a logarithmic scale.

Figure 5 presents the root-mean-square error $\sigma_{RF}$. It was also averaged for different (against the trend minimum) wave packet position. Figure 5 shows that the cubic smoothing spline possesses the least standard deviation of all the filtration techniques presented. One may highlight a group of filters with good results: CMA filter, smoothing cubic spline, Hodrick–Prescott Filter, L1 Filter, double CMA filter. The smoothing cubic spline is the best among them; it yields the results that are, on average, by a factor of 1.5–2 better, than those of the remaining in the above group.

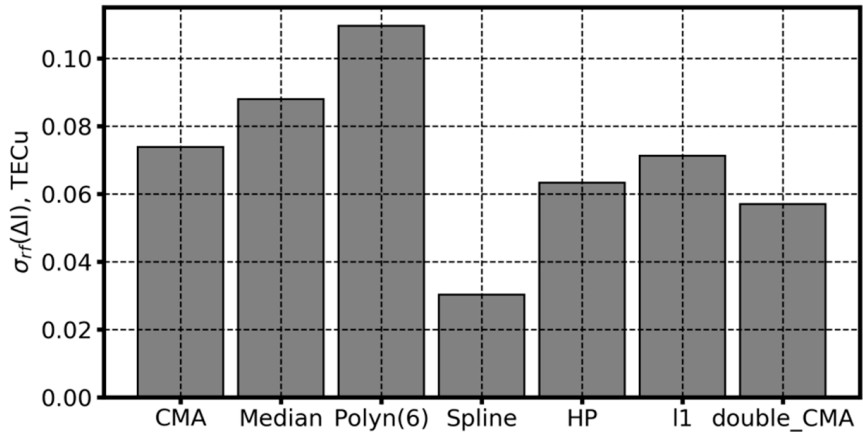

**Figure 5.** Root-mean square error, $\sigma_{RF}$, for various detrending techniques.

Figure 6 shows the dependence of the correlation coefficient between the modeled and the recovered signals on the trend velocity in the wave-packet location. One can see that the cubic smoothing spline possesses the highest correlation coefficient, i.e., this corresponds to the best detrending (by this criterion).

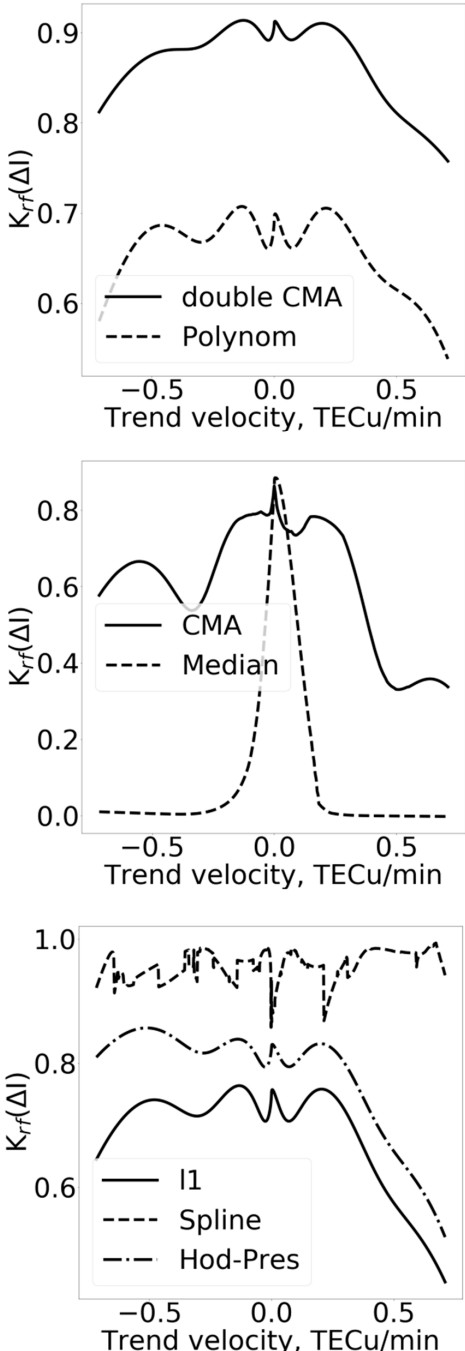

**Figure 6.** Trend velocity vs. the correlation coefficient.

It is interesting that double CMA yields better results than CMA. We may elucidate this as follows: The trend is also a variation but of a significantly larger period. Herewith, in the addressed example, the trend varies from 62 to 0 TECu and back, whereas the wave-packet high-frequency variations vary within –0.2 and 0.2 TECu. That is, the trend-related variations have the amplitude hundredfold larger, than that for the wave-packet variations. Because the moving average filter has an amplitude-frequency characteristic that is sufficiently smooth at the boundaries, the trend as a slow variation permeates into the output signal. In this case, its amplitude becomes essentially lower owing to the filter suppression. Applying the moving average filter anew removes the trend variation odds from the output signal.

Previous consideration takes into account one and the same trend. To analyze more realistic situation we used modeling based on the International Reference Ionosphere. We used IRI-2012 [18]

with URSI coefficients and NeQuick topside options. The total number of TEC IRI series was 6144. Figure 7 shows the distribution of error of detrending based on the 6th-order polynomial and the cubic smoothing spline showing best performance. Mean bias error and root-mean square error clearly show advantage of spline detrending over polynomial those. At least spline does not introduce RMSE more than 0.25 TECU and mean bias error > 0.4 TECU for IRI model trends estimating, while for 6th-order polynomial these errors can exceed 1 TECU in some cases. However, in 95% the RMSE and MBE does not exceed 0.05 TECU.

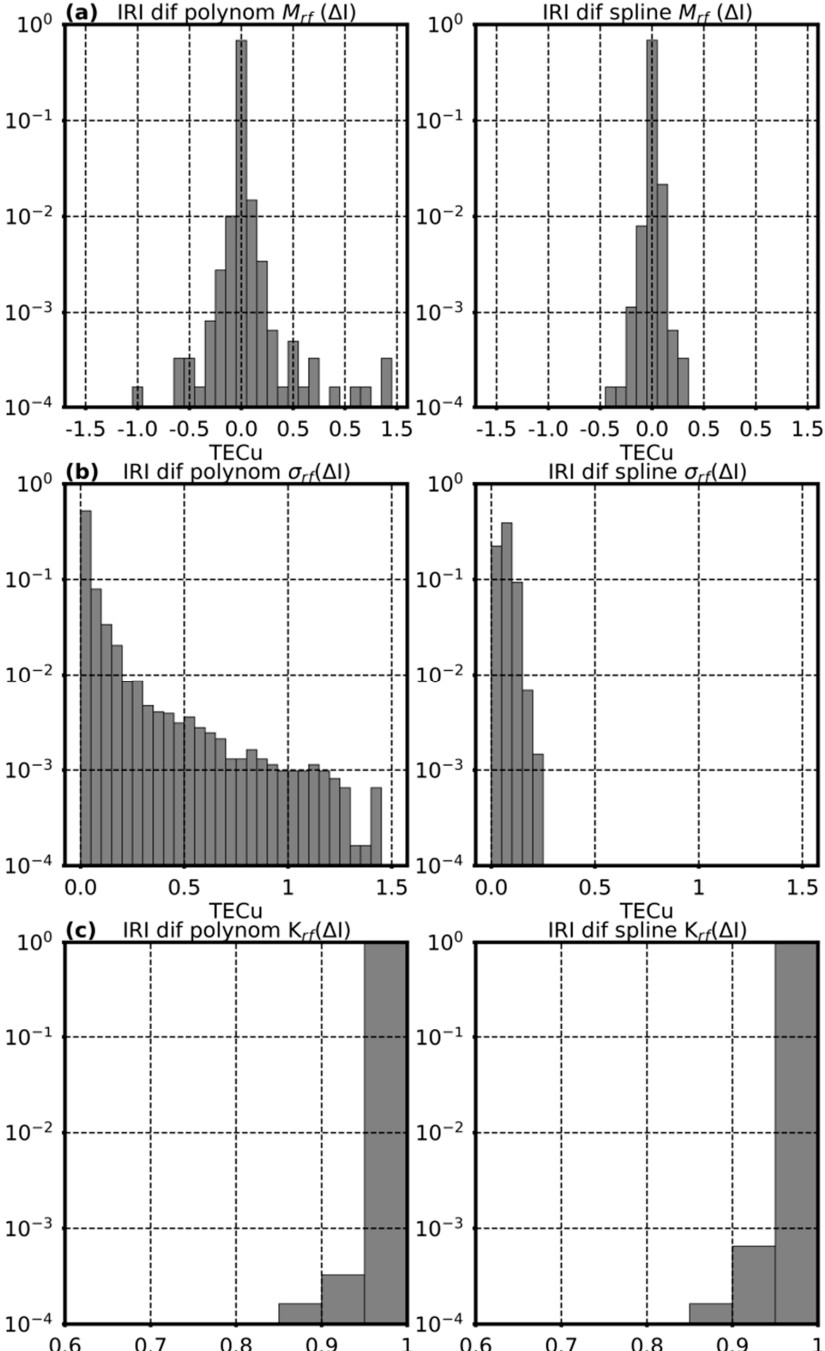

**Figure 7.** The distribution of trend IRI TEC error: (**a**) Mean bias error- left for polynomial, right for spline; (**b**) Root-mean-square error—left for polynomial, right for spline; (**c**) Correlation coefficient—left for polynomial, right for spline.

Summarizing the results, we can state that the best detrending algorithm for TEC data is the smoothing cubic spline.

### 3.2. Variation Selection

To solve the variation selection problem, we chose the following approach. We generated a signal (wave packet) comprising the harmonics that are inside all the ranges that we are interested in, and beyond (T < 2 min, 2–10 min, 10–20 min, 20–60 min, > 60 min). We included the harmonics with the periods of 1 min 40 s, 2 min 30 s, 10 min, 15 min, 25 min, 40 min, 51 min 40 s, 108 min 40 s. The amplitude of each harmonic is 0.2 TECU. The mean bias error, RMSE and correlation coefficient should depend on the ratio of the amplitudes of the modeled signal and noise. Therefore, we added Gaussian noise with different amplitudes to the signal. As the "correct" recovered signal, we accepted the harmonic in the wave packet (4), corresponding to the filtering bands (2–10 min, 10–20 min, 20–60 min) and that were introduced above. Here with we check "preserve relative amplitudes of spectral components in the filtered band" and "preserve spectral composition only for the filtered band" conditions.

Figure 8 illustrates the modulated signal: top, in the frequency domain (the oscillation period is on the abscissa axis) and, bottom, in the temporal domain.

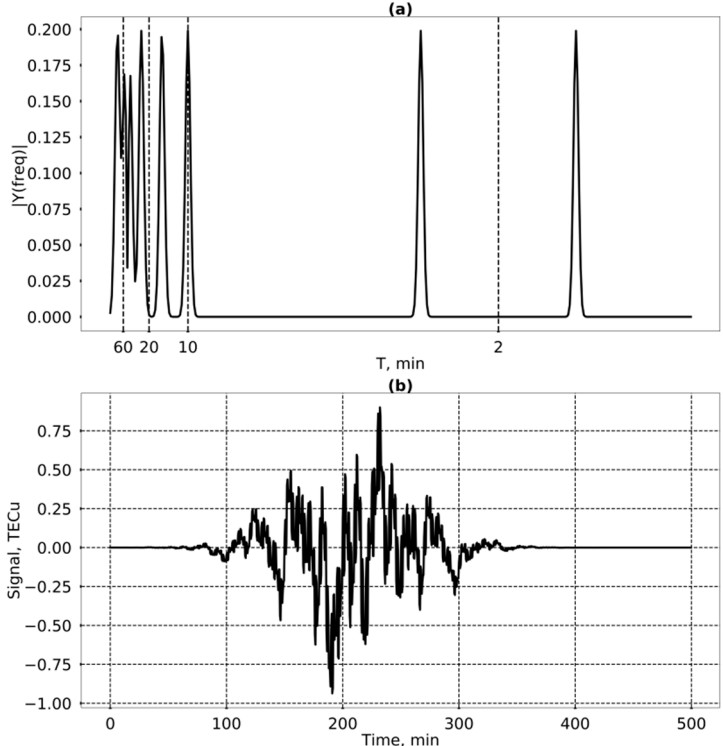

**Figure 8.** Modeled signal in the frequency (**a**) and temporal (**b**) domains.

Figure 9 shows the variation selection quality dependence at various levels of the additive noise. The top panel exhibits the dependence of the mean bias error $M_{RF}$ on the signal-to-noise ratio, middle panel; the root-mean-square error $\sigma_{RF}$ dependence, bottom panel; and the correlation coefficient $K_{RF}$ dependence. In general, the signal-to-noise ratio decrease leads to decrease in the filtering quality. By the criteria of $K_{RF}$, the moving average and the Butterworth filter yield the best results, and by the $M_{RF}$ criterion, the best results are from CMA. Both the Butterworth and CMA filters showed minimal RMSE.

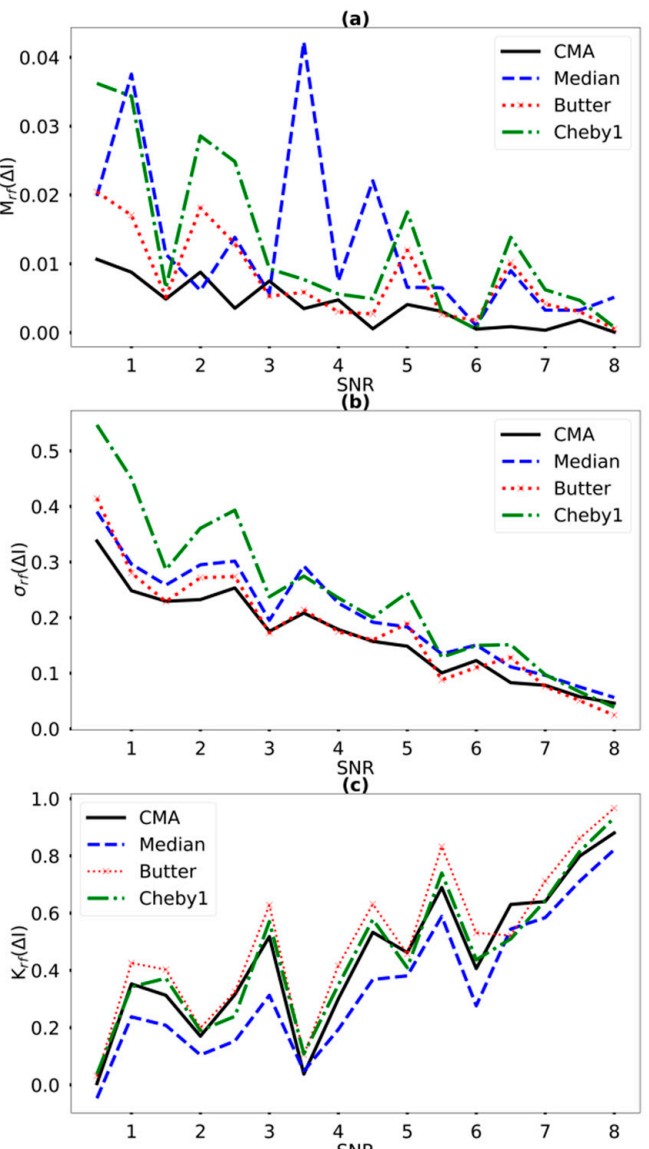

**Figure 9.** Dependence of the (**a**) $M_{RF}$, (**b**) $\sigma_{RF}$, (**c**) $K_{RF}$ selection quality criteria on the white-noise amplitude.

## 4. Experimental Results

To realize how dramatically the results from various techniques may differ, when working with real experimental data, we processed the UZUR receiver data. The receiver is located within the ISTP SB RAS polygon in the settlement of Uzur [19]. We used data on 14 April 2016. Figures 10 and 11 present the results for the "UZUR - GPS26" lines-of-sight.

Figure 10 shows the experimental TEC series and the trend based on the smoothing cubic spline and the 6th-order polynomial. Generally, one can see that the spline enables to obtain a reasonable trend. Note that the smoothing cubic spline operates correctly, even in the loss-of-lock cases. If a loss-of-lock occurs, the cubic smoothing spline forms independent trends for parts.

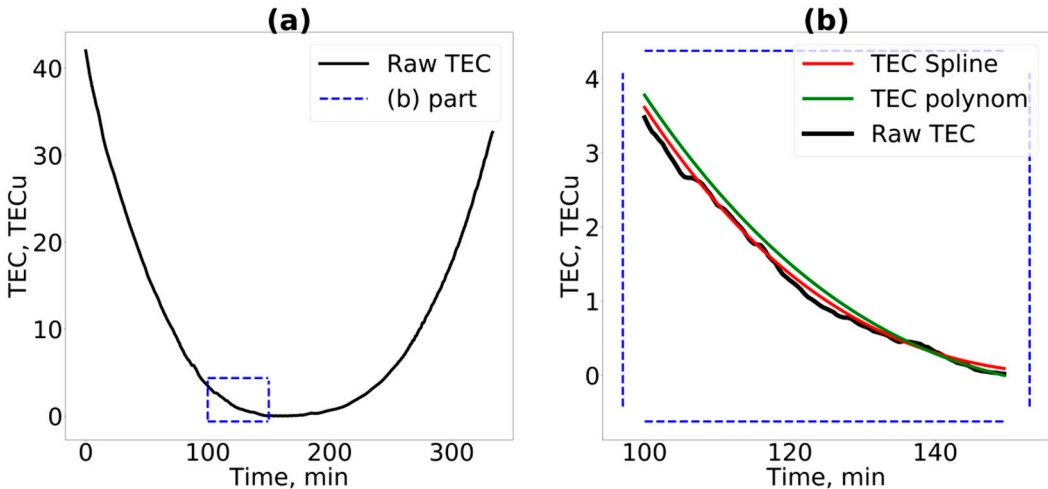

**Figure 10.** Raw TEC (**a**) and detailed part of Raw TEC (**b**). On the right panel red and green lines show TEC trends based on smoothing spline and polynomial, correspondingly Data from "UZUR - GPS26" on 14 April 2016.

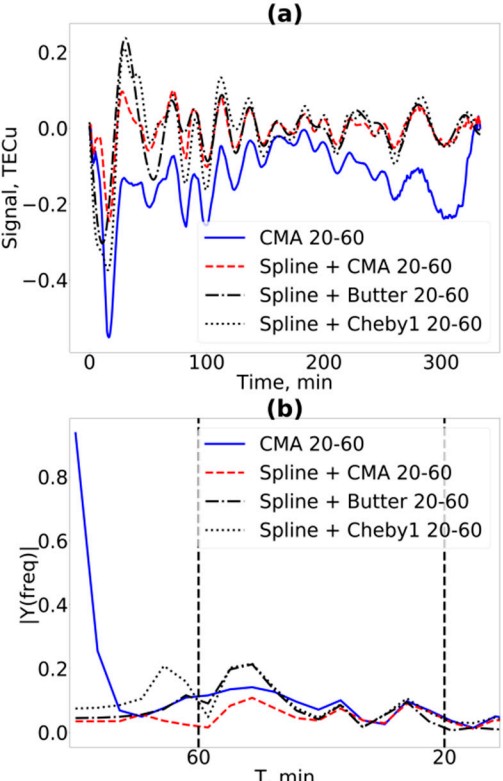

**Figure 11.** TEC variations obtained upon detrending and filtering through various techniques: (**a**) in the temporal domain; (**b**) in the frequency domain. The solid line corresponds to the CMA filtering (with a 20–60 min window). The other three cases correspond to, spline detrended TEC series with filtering based on the CMA (dotted line), Butterworth filter (dash-dotted line), and Chebyshev filter (dotted curve). Data from "UZUR - GPS26".

Figure 11 (top panel) shows filtering the 20–60 min variations. The solid line corresponds to CMA filtering (with a 20–60 min window). In the other three cases, we, first, removed the TEC trend, based on the spline, and then filtered, based on the CMA (dotted line), Butterworth filter (dash-dotted line),

and Chebyshev filter (dotted curve). The bottom panel shows the spectra of the corresponding signals. The least satisfactory results correspond to using the moving average without preliminary detrending. The spectrum demonstrates the presence of considerably constant component, and, in the temporal dynamics, we see only negative values. Detrending through the smoothing cubic spline removes this artifact.

Using the centered moving average for filtering shows the best results: the signal view and its frequency composition in the set frequency range persist. These results correlate well with the predictions from the model section. Processing artifacts that occur during variations filtering are also clearly visible.

## 5. Discussion

Modeling based on simple analytical defined trend showed, that errors introduced by various filtering procedures are quite different. The worst mean bias error was found for CMA and Median detrending. In terms of statistics, CMA detrending is optimal for normal distribution of fluctuation around the trend, while Median is optimal for Laplace distribution of those [20]. It assumes trend stability. CMA filtering cannot easily be generalized to remove trends [21]. Therefore, it is TEC trend nonlinearity, which does not allow to effectively use these two filters. The other (cubic spline, Hodrick–Prescott filter, L1 filter, 6th-order polynomial) detrending procedure, which we used, better fit the trend nonlinearity. Therefore, they produce smaller mean bias error under detrending (see Figure 4). However, Hodrick–Prescott filter and L1 filter resulted in relatively small (0.6–0.85) correlation coefficient as compared to cubic spline. At first glance, the 6th-order polynomial should reproduce used trend (5) in perfect way. However, the modulo operator results in nonlinearity of higher (than six) orders. The Hodrick–Prescott filter and L1 filter most probably overfit the trend to reduce variations in the real signal. The latter decreases correlation. We should note that the Hodrick–Prescott filter and L1 in some cases can be improved by an adjustment of parameters. However, it is difficult to adjust them for all the cases or make an adaptive procedure.

Detrending of the realistic IRI trends also showed the appearance of the MBE. The IRI trend is more nonlinear than those described by Equation (5). Nevertheless, under spline detrending, in 95% the MBE did not exceed 0.05 TECU. Due to super fountain effect, the TEC in the equatorial anomaly region can drastically change and reaches 170–180 TECU during strong magnetic storms [22]. Spatial-time TEC gradients during the storm can amplify nonlinearity and thus increase the MBE and decrease correlation.

To analyze how significant detrending influences the results, we also carried out statistical analysis. We treated TEC series in two different ways. The first way includes spline detrending and CMA filtering ($dTEC_{detr}$), while the second one includes only CMA filtering ($dTEC_{nodetr}$). We used GPS/GLONASS data from 3532 stations from the SIMuRG [15] database for 21 February 2019. The total number of TEC series was 169,731. Figure 12 shows the distributions of difference $dTEC_{detr}$-$dTEC_{nodetr}$ for all three studied variation periods. The distributions were normalized by the total number of points in the distribution. The results show that the larger periods show higher errors. Similar results were obtained in the modeling part. Thus, the probability of error increases with increasing periods of variation. The detrending procedure is most crucial for filtering the variations with longer periods.

The main conclusion to make from Figure 12 is that detrending is actually a necessary procedure for TEC data treatment: an absolute error of >0.1 TECU will have >1%, 1%, 20% measurements for 2–10 min, 10–20 min, and 20–60 min TEC variations, correspondingly.

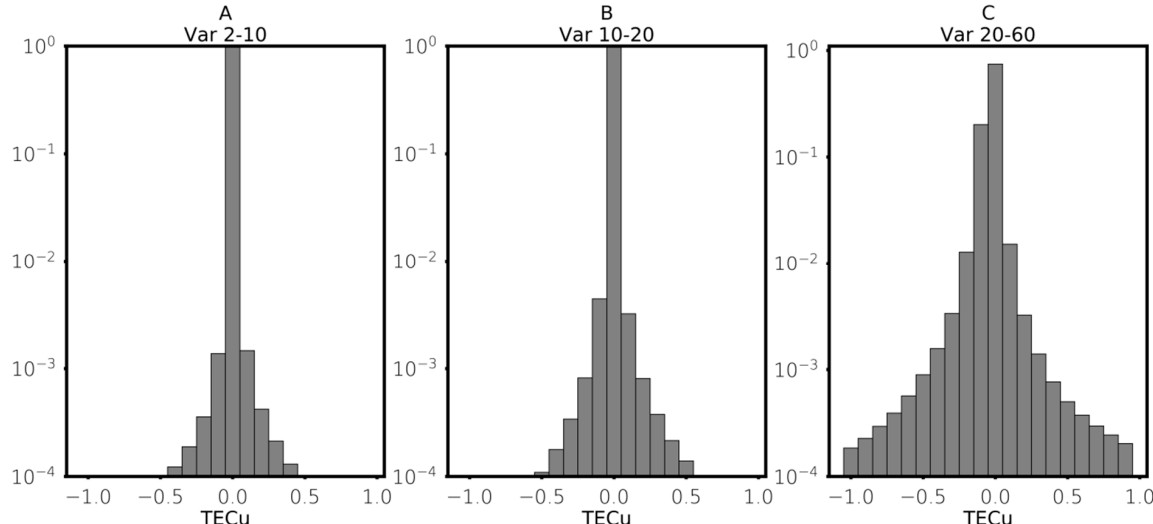

**Figure 12.** Detrending influence. Distributions of the difference between the two data processing options for the TEC variations of 2–10 min (**a**), 10–20 min (**b**), and 20–60 (**c**). The y-axis is in logarithmic scale.

CMA and the Butterworth filters yield the best results for variation selection. All filters were designed for different applications. Butterworth filters were suggested for applications where maximum pass band flatness is required ("flat maximally magnitude" filters) [23], while Chebyshev filters are for rapid transition from passband to stopband. On the one hand, the CMA filter is the easiest digital filter to use, while on the other it is optimal for reducing white random noise keeping a sharp transition from passband to stopband [24], while the median filter is for Laplace noise. Probably, because it is white noise which we used for modelling, the median filter yields maximal errors. However, we expect just the Gaussian white noise, which is typical for GNSS measurements [25] (except for strong scintillation conditions).

Another important issue is Gibbs' phenomenon. If the transition from passband to stopband is steep there can be leakage of unwanted energy into the filtered signal and ringing effects. For the Butterworth filters Gibbs' phenomenon can take place [26]. From this, it follows that the moving average is one of the best selection versions.

The obtained information on the variations in different ranges may be used to study diverse processes in the ionosphere, including investigations in the automatic mode, for example, for detecting solar flares [27,28]. Reducing the number of errors related to data processing itself and not linked to the physical process is very topical.

## 6. Conclusions

We checked the centered moving average, centered moving median, 6th-order polynomial, Hodrick–Prescott filter, L1 filter, cubic smoothing spline, double application of the moving average for GNSS-TEC detrending, and centered moving average, centered moving median, Butterworth filter, type I Chebyshev filter for TEC variation selection.

Our modeling enables to draw the following conclusions:

1. The problem of the GNSS data filtering should be split into two subtasks (detrending and selection) to obtain the best results. At such an approach, the possibility to obtain artifacts related to the data processing algorithms is minimal.
2. Results show that the longer periods, the higher errors can appear caused by insufficient detrending. For the detrending problem, the smoothing cubic spline provides the best results among the versions presented in this paper. It features the minimal value of the mean bias error and the root-mean-square error, as well as the maximal correlation coefficient. The 6th-order polynomial also produces good results and can be used for this task.

3.   For the filtering problem, the centered moving average filter showed the best results among the versions presented in this paper.

We are planning to incorporate above suggestion to systems for automatic treatment, like SIMuRG (https://simurg.iszf.irk.ru) [15].

**Author Contributions:** Conceptualization: B.M. and Y.Y.; methodology: B.M., Y.Y., and A.M.; investigation: B.M., Y.Y., and A.V.; software: B.M.; resources: Y.Y. and A.V.; formal analysis: B.M.; writing—original draft preparation: B.M. and Y.Y.; writing—review and editing: B.M., Y.Y., and A.V.; visualization: B.M.; funding acquisition: Y.Y. All authors have read and agreed to the published version of the manuscript.

**Funding:** This study was funded by the Russian Science Foundation, grant no. 17-77-20005.

**Acknowledgments:** The authors thank Denis Sidorov for valuable discussion. In the research, we used experimental data from IGS [16] and from SibNet [19] that is a part of the equipment of Center for Common Use «Angara» (http://ckp-rf.ru/ckp/3056/) operating under budgetary funding from Basic Research Program II.16. One can access used experimental data from https://simurg.iszf.irk.ru.

**Conflicts of Interest:** The authors declare no conflict of interest.

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
