# Peer review of "Wave Signatures in Total Electron Content Variations: Filtering Problems"

_remotesensing, doi:10.3390/rs12081340_

Round 1

Reviewer 1 Report

It would be useful to add two more fragments in Fig. 1: a - the original data of these stations; and b - detrended ones without filtering (or only filtered without detrending, if it looks better).

Line 132 “a trend, a signal” May be: “a trend and a signal”?

Line 146  Not “Where”, but “where”

Line 148-149 The values of B and t_0 are not defined. How this cubic function corresponds to Fig. 2 b?

Fig. 10 must be described better. There are 4 types of lines. It seems that a fragment is separated and increased. Corresponding geometrical lines must differ from “TEC spline”. Is thin solid line the same as thick one? Does the black dashed line mean absence of data? It would be good to show the results of some other methods just here for demonstrative comparison, if the difference can be seen. 

Fig 12. It would be better to change indexes A,B in dTEC_A, dTEC_B to avoid similarity to A,B,C panels.

Author Response

We are thankful to the reviewer for his/her useful comments which allow us to improve the article as well as for interest to the article and immediate response.  All issues were addressed in the manuscript.

  • It would be useful to add two more fragments in Fig. 1: a - the original data of these stations; and b - detrended ones without filtering (or only filtered without detrending, if it looks better).

We added 4 new panels in Fig. 1: 

(a) Slant TEC and TEC trend obtained by 60-min centered moving average for IRKJ station. 

(b) Slant TEC and TEC trend obtained by 60-min centred moving average for BADG station.

(c) TEC series detrended by 60-min central moving average for IRKJ station 

(d) TEC series detrended by 60-min central moving average for BADG station

All original TEC series are from G06 satellite signal.

  • Line 132 “a trend, a signal” May be: “a trend and a signal”?

Done

  • Line 146  Not “Where”, but “where”

Done

  • Line 148-149 The values of B and t_0 are not defined. How this cubic function corresponds to Fig. 2 b?

We agree with reviewer and added definition of B and t_0 and its values. The t0 parameter determines the position of the trend minimum and is equal 250 minutes (half of the array length), B is amplitude of the trend and equal 62/(2503).

  • Fig. 10 must be described better. There are 4 types of lines. It seems that a fragment is separated and increased. Corresponding geometrical lines must differ from “TEC spline”. Is thin solid line the same as thick one? Does the black dashed line mean absence of data? It would be good to show the results of some other methods just here for demonstrative comparison, if the difference can be seen. 

We appreciate the comments. Fig. 10 was really described in a bad way. We changed the figure for better susceptibility and accessibility of information. Now corresponding geometrical lines differ from “TEC spline” by color and type of lines, also we added polynomial-based trend to show the results of some other methods.  

  • Fig 12. It would be better to change indexes A,B in dTEC_A, dTEC_B to avoid similarity to A,B,C panels.

We suggest to to change indexes “A”,”B” in dTEC_A, dTEC_B to “detr” (with detrending) and “nodetr”(no detrending). 

Also we would like to note, that we changed structure of manuscript due to journal sections policy. We added a self stand Discussion part before Conclusions

Reviewer 2 Report

The paper is well written and well organized. It is important to state that the comparison of various filtering/smoothing/detrending techniques were already performed by many authors especially in the field of signal processing. However it is interesting to read about this in the context of TEC variations. I would suggest removing the part about the performance. The description of CPU used (Core i7) is ambiguous since these processors are available on the market for many years in many versions. Also implementation affects the computational cost significantly and there is no information about it. In my opinion it is easier (and with a very little loss to the paper) to remove this information than to add implementation details.

Author Response

We are thankful to the reviewer for his/her useful comments which allow us to improve the article as well as for interest to the article and immediate response. All issues were addressed in the manuscript.

  • I would suggest removing the part about the performance. The description of CPU used (Core i7) is ambiguous since these processors are available on the market for many years in many versions. Also implementation affects the computational cost significantly and there is no information about it. In my opinion it is easier (and with a very little loss to the paper) to remove this information than to add implementation details.

We agree that the results of this manuscript part may depend on a large number of factors, for example, the quality of writing the code and its structure in addition to the characteristics of a computer. Adding the source code of one or another filter can solve this problem, but it will clutter up the manuscript itself and distract from the main idea. We suggest removing this part.

Also we would like to note, that we changed structure of manuscript due to journal sections policy. We added a self stand Discussion part before Conclusions